# Genome Size Estimation of *Callipogon relictus* Semenov (Coleoptera: Cerambycidae), an Endangered Species and a Korea Natural Monument

**DOI:** 10.3390/insects12020111

**Published:** 2021-01-27

**Authors:** Yun-Sang Yu, Soyeong Jin, Namjoon Cho, Jongok Lim, Cheol-Hak Kim, Seung-Gyu Lee, Sangil Kim, Jong-Seok Park, Keekwang Kim, Chungoo Park, Sung-Jin Cho

**Affiliations:** 1School of Biological Sciences, College of Natural Sciences, Chungbuk National University, Cheongju 28644, Korea; yys9002@chungbuk.ac.kr (Y.-S.Y.); jpark16@chungbuk.ac.kr (J.-S.P.); 2School of Biological Sciences and Technology, Chonnam National University, Gwangju 61186, Korea; owljin.sy@gmail.com; 3Department of Biochemistry, Chungnam National University, Daejeon 34134, Korea; skawns119@naver.com; 4Division of Forest Biodiversity, Korea National Arboretum, Pocheon 11186, Korea; jolim79@korea.kr; 5Institution of Biological Resources, Osang K-Insect, Yesan 32426, Korea; petass@empas.com; 6Animal Resources Division, National Institute of Biological Resources, Incheon 22689, Korea; jspdi84@korea.kr; 7Museum of Comparative Zoology and Department of Organismic and Evolutionary Biology, Harvard University, Cambridge, MA 02138, USA; sikim@g.harvard.edu

**Keywords:** *Callipogon relictus*, endangered species, flow cytometry, genome size, *k*-mer analysis, longhorned beetle

## Abstract

**Simple Summary:**

The longhorned beetle *Calipogon relictus* has been considered as a class I endangered species since 2012 in Korea. In an attempt towards beetle conservation, we estimated its genome size at 1.8 ± 0.2 Gb, representing one of the largest cerambycid genomes. This study provides useful insight at the genome level and facilitates the development of an effective conservation strategy.

**Abstract:**

We estimated the genome size of a relict longhorn beetle, *Callipogon relictus* Semenov (Cerambycidae: Prioninae)—the Korean natural monument no. 218 and a Class I endangered species—using a combination of flow cytometry and *k*-mer analysis. The two independent methods enabled accurate estimation of the genome size in Cerambycidae for the first time. The genome size of *C. relictus* was 1.8 ± 0.2 Gb, representing one of the largest cerambycid genomes studied to date. An accurate estimation of genome size of a critically endangered longhorned beetle is a major milestone in our understanding and characterization of the *C. relictus* genome. Ultimately, the findings provide useful insight into insect genomics and genome size evolution, particularly among beetles.

## 1. Introduction

The longhorned beetle genus *Callipogon* Audinet-Serville, 1832 [1] (Coleoptera: Cerambycidae) consists of five subgenera including nine species worldwide. Only a single species is found in East Asia (Korea, China, Far Eastern Russia), while the remaining species are distributed across Central and South America, including Mexico, Guatemala, and Colombia [2,3]. The relict longhorn beetle *Callipogon relictus* Semenov, 1898 [4] is the sole Asian representative of the genus, and thus represents one of most intriguing insects in the Palearctic region, both in terms of its biogeographical and ecological history, as well as its unequivocal importance for conservation. As such, *C. relictus* is strictly protected in Korea under the national natural monument no. 218 since 1968, and as a class I endangered species since 2012. *C. relictus* in East Asia has been suggested to represent a biogeographical link between the faunas of the Old World and New World when the Bering land bridge was exposed above the sea level [5,6].

*C. relictus* requires five to six years to complete the life cycle under natural conditions [7]. The host plant records suggest that *C. relictus* is polyphagous, feeding on 17 different species of broadleaved trees belonging to seven families [8]. In particular, *C. relictus* larvae were found to feed mainly on *Quercus* spp. and *Carpinus laxiflora* in the Gwangneung Forest, Korea, based on the investigation conducted by the Korea National Arboretum, and on *Ulmus davidiana* var. japonica in the Ussuri Nature Reserve, Russia (Kuprin, A.V., pers. comm.). This exceptionally wide host range is plausible given that *C. relictus* is primarily fungivorous, deriving nutrients during larval development from fungal mycelia in decaying wood, similar to most other saproxylophagous beetles [9].

To date, a total 138 insect genomes have been sequenced. However, only eight of them represent Coleoptera, most of which are regarded as important insect pests, including *Agrilus planipennis* (Buprestidae); *Anoplophora glabripennis* (Cerambycidae); *Dendroctonus ponderosae* and *Hypothenumus hampei* (Curculionidae); *Leptinotarsa decemlineata* (Chrysomelidae); and *Tribolium castaneum* (Tenebrionidae) [10]. McKenna et al. (2016) alone published the entire genome sequence of a longhorn beetle—*A. glabripennis*—with a genome size ranging between 981 and 970 Mb in female and male individuals, respectively. Based on the comparative genomic analyses, McKenna et al. (2016) concluded that the expansion and functional differentiation of the genes associated with specialized plant feeding facilitated the adaptation of *A. glabripennis* to a variety of new host plants in its new habitat [11].

The two common approaches for estimating genome size include flow cytometry and *k*-mer analysis. Flow cytometry is a fluorescence-based technique used to detects the intensity of fluorescence emitted by DNA stained with propidium iodide [12]. As a relatively quick and reliable method for accurately estimate the size of even large genomes, flow cytometry has been widely used to analyze various insect genomes, such as in firefly [13], the stick insect *Clitarchus hookeri* [14], Neotropical mutualistic ant [15], and *Helicoverpa* moths [16]. Nevertheless, the application of flow cytometry is limited by the availability of intact tissue [17] and the estimate is also affected by chromatin condensation and the proportion of cells in G_0_ to G_1_ phases. Given that insect tissues may show high levels of endoreplication, the use of appropriate tissue for the analysis and selection of proper standard species with well-known genome size are critical for accurate size estimation using flow cytometry [16,18].

However, *k*-mer analysis entails sequence-based estimation utilizing high-throughput sequencing data, and therefore, is independent of the stage of cell cycle, as well as the integrity of the tissue used. This method also facilitates measurement of genome properties, such as the rate of heterozygosity [19,20]. Nonetheless, *k*-mer based estimates alone are easily affected by repetitive element in the genome [21,22,23], and may result in underestimation of genome size [24]. Given the apparent caveat, *k*-mer approach has often been used in conjunction with flow cytometry, particularly in studies involved de novo assembly of arthropod genomes (e.g., the spider *Dysdera silvatica* [23] and caddisflies [22]).

In this study, we employed both flow cytometry and *k*-mer analysis to deduce the genome size of the critically endangered relict longhorn beetle, *C. relictus*. As the initial step towards expanding on the studies of longhorn beetle genomics, two independent approaches were concurrently used to estimate the genome size. The size estimate of the *C. relictus* genome was larger than that of most of the other beetle genomes assembled to date (e.g., 1.17 Gbp for the Colorado potato beetle, *Leptinotarsa decemlineata* [25]). We discuss its implications for studies investigating cerambycid genomics.

## 2. Materials and Methods

### 2.1. Sample Preparation

The *Callipogon relictus* specimens used in the present study include the second-generation offspring of the beetle collected from the Korea National Arboretum on 20 July 2017. The larvae were reared on a fungal diet and under the 14L:10D (14 h light:10 h darkness) photo-period at 24 ± 1 °C and RH of 60% or 65% for adult. We extracted genomic DNA from leg muscle of an unmated one female adult, 2 weeks after eclosion using MagAttract HMW DNA kit (Qiagen, catalog no.67563) according to the manufacturer’s instructions. Final genomic DNA was eluted in 100 µL of Solution AE.

### 2.2. Genome Size Estimation by Flow Cytometry

The whole tissue samples except the internal organs of *C. relictus* one larvae were dissected to estimate the genome size using flow cytometry. Ten-month-old male C57BL/6J mouse liver tissues were dissected and used as a control. Dissected tissues were digested with 1 mg/mL collagenase/dispase (Sigma-Aldrich, 10269638001, St. Louis, MO, USA) at 37 °C for 1 h, followed by trypsinization and filtering with 70 µm cell strainers (SPL, Pocheon, South Korea) to isolate single cells. The cells were then fixed in cold 70% ethanol overnight, stained with 50 µg/mL of propidium iodide (Sigma-Aldrich, USA), and treated with 125 µg/mL of RNase A (iNtRON, DaeJeon, South Korea). The relative size of genomic DNA in *C. relictus* and mouse was analyzed with FlowJo (TreeStar, San Jose, CA, USA) based on the fluorescence intensity using a flow cytometry (BD Bioscience, San Jose, CA, USA).

Geometric log mean values were used as the mean fluorescence intensity (MFI) to calculate the genome size of *C. relictus*, using the formula below based on the comparison with the MFI of mouse, *Mus musculus,* whose genome size is 2.67 Gb. Each MFI value was determined from three independent experiments.

Genome size of *C. relictus* (bp) = (*G*_0_/*G*_1_ peak MFI of *C. relictus*)/(*G*_0_/*G*_1_ peak MFI of *M. musculus*) × genome size of *M. musculus* (bp).

### 2.3. Genome Size Estimation by k-mer Analysis

Genomic DNA library was prepared with a Truseq Nano DNA Prep Kit (Illumina, San Diego, CA, USA) by first randomly shearing 200 ng of genomic DNA into 550 bp inserts using the Covaris S2 system (Covaris, Woburn, MA, USA). Next, a single ‘A’ nucleotide was added to the 3′ blunt-ends of fragmented DNA and the adapters were ligated to both ends of the fragmented DNA. The adapter-ligated DNA was PCR amplified to increase the concentration of the ligated DNA fragments. Bioanalyzer (Agilent, Santa Clara, CA, USA) was used to verify the length distribution of the amplified library. In addition, qPCR was performed to quantify the final library using SYBR Green PCR Master Mix (Applied Biosystems, Foster City, CA, USA). Finally, the verified library was sequenced using paired-end 101 bp reads on the Illumina NovaSeq6000 flow cell platform (Illumina, San Diego, CA, USA). As a result, a total of 60 Gb of raw sequence reads were generated (project accession PRJNA689978).

Prior to *k*-mer analysis, all raw sequence reads were pre-processed by Trimmomatic (v0.39) [26] to trim adapter sequences and eliminate low quality reads. Using the trimmed reads, *k*-mer analysis was performed to estimate the genome size. The *k*-mer frequency distributions with the values of *k* ranging between 17 and 23 bp with a 2-bp interval were estimated using Jellyfish (v2.3.0) [27]. The final genome size was calculated by dividing the total number of *k*-mer by the peak value of *k*-mer frequency distribution. Additionally, GenomeScope (v1.0) [28] was used to characterize the genome of *C. relictus*, including genome size, rates of heterozygosity, and repeat content.

## 3. Results

*Callipogon relictus* is shown in Figure 1. To estimate the genome size of *C. relictus*, we first quantified DNA contents of *C. relictus* larval cells and mouse liver cells using flow cytometry.

Based on the difference in fluorescence intensity between the two organisms, the *C. relictus* genome was smaller than that of mouse. Based on the known mouse genome size of 2.67 Gb, we inferred the size of *C. relictus* genome at about 2.00 Gb, using the MFI of mouse liver cells as a reference (Figure 2).

We generated high-throughput genomic sequence data from the leg muscle of a female adult specimen to estimate sequence-based genome size by *k*-mer analysis. The generated raw sequence reads and trimmed reads are presented in Table 1. Based on two different modes of *k*-mer analyses, we estimated the genome size of *C. relictus* to range from 1,517,383,829 bp to 1,882,948,731 bp, as summarized in Table 2.

The distributions of *k*-mer coverages based on Jellyfish analysis presented double peaks, with the heterozygous peak recovered at coverage 11 (21-mer and 23-mer) and 12 (17-mer and 19-mer) and the homozygous peak at coverage 20 (23-mer), coverage 21 (21-mer), coverage 22 (19-mer), and coverage 23 (17-mer) (Figure 3).

In addition, based on the distribution of *k*-mer frequency, we evaluated the properties of the *C. relictus* genome using GenomeScope, which yielded an estimated heterozygosity rate of 1.70–1.81% and an estimated repeat length of 757–1096 Mb (Figure 4).

## 4. Discussion

This study represents the first attempt to estimate genome size within the family Cerambycidae by *k*-mer analysis. Because of possible endoreplication in insect cells and tissues, the use of flow cytometry alone may result in inaccurate estimation of genome size depending on the type and stage of the tissue used for the analysis [18]. Therefore, it is important to complement flow cytometry-based estimates by another independent method, such as *k*-mer analysis. The Illumina sequence reads generated for *k*-mer analysis can be further used to directly assemble the whole genome sequence *de novo*.

The genome size of 1.8 ± 0.2 Gb represents one of the largest longhorn beetle genomes reported to date, and is more than twice the size of the Asian longhorn beetle (*Anoplophora glabripennis*) genome. However, this result is not surprising given the apparent genome size variation reported previously within the family Cerambycidae, ranging from 528 Mb for the read-headed ash borer, *Neoclytus acuminatus* (subfamily Cerambycinae) to 1.88 Gb for the live-oak root borer, *Archodontes melanopus* (subfamily Prioninae), although these results were based solely on flow cytometry analysis [17]. The genome size varies across insects even more remarkably, with the largest insect genome discovered in the mountain grasshopper *Podisma pedestris* (1C-value = 16.93 pg) nearly 250-fold larger than the smallest genome of the non-biting midge *Clunio tsushimensis* (1C-value = 0.07 pg) [29,30,31]. Given the significant variation in size across insects and even among longhorn beetles, we expect the current findings to contribute to the burgeoning amount of insect genome data. As further genomic data become available across diverse insect lineages, we may conduct comparative genomic analyses to delineate the genetic mechanism underlying the evolution of various ecological and physiological traits of insects, such as immune system, metabolic detoxification, parasitism and polyphagy [11].

Finally, given the importance of conservation, the genomic study of this critically endangered longhorn beetle is expected to offer useful information for developing an effective conservation strategy. Nevertheless, only a handful number of studies reported the molecular analysis of *C. relictus* in Korea based on sequencing of the Cytochrome c oxidase subunit I (COI) barcode gene from a cerambycid larval species collected and identified from the Gwangneung Forest [32]. The complete mitochondrial genome sequence of *Callipogon relictus* has been published [33]. Furthermore, the phylogenetic and biogeographic history of *C. relictus* based on multilocus sequence data has been obtained from multiple geographical populations of *C. relictus*, together with most of its congeners worldwide [5].

## 5. Conclusions

The current findings represent a pioneering effort in the study of *Callipogon relictus* evolutionary genomics. Additionally, comparative genomic studies in the future are expected to enable conservation efforts based on key loci that are, contributing to inbreeding depression and disease susceptibility, as well as the fitness of potential introgression [34].

## Figures and Tables

**Figure 1 insects-12-00111-f001:**
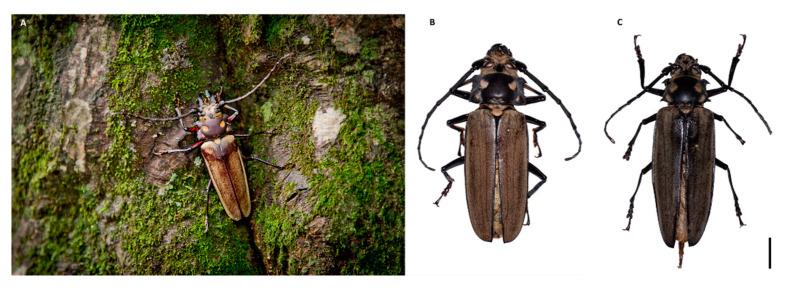
(**A**) Living specimen in the Gwangneung Forest. (**B**,**C**) Dorsal aspect of male and female. Scale bars: 10 mm.

**Figure 2 insects-12-00111-f002:**
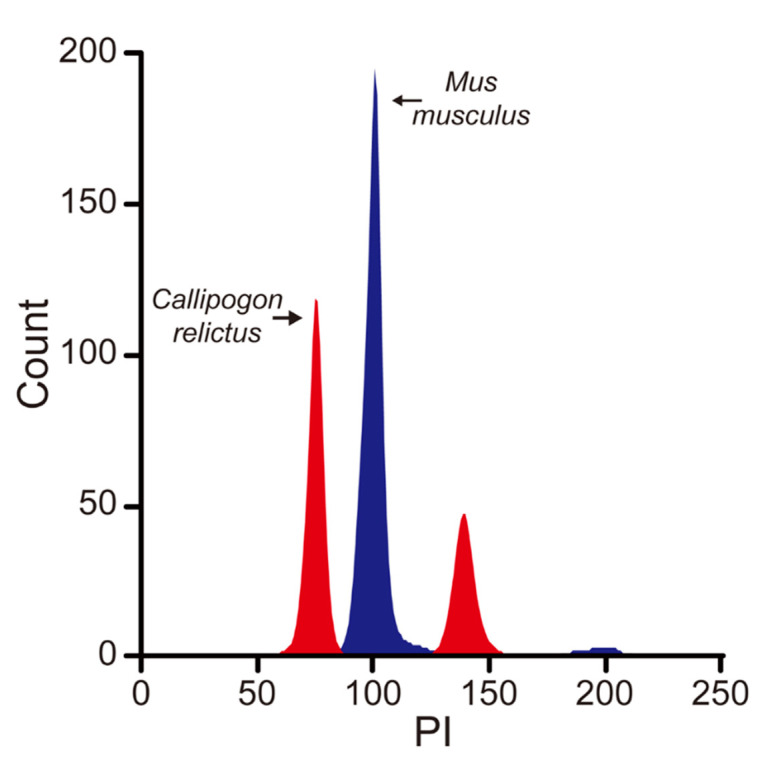
*C. relictus* larval cells are stained with propidium iodide and subjected to flow cytometry analysis. Mouse liver cells were used as an internal standard to evaluate the genome size of *C. relictus*.

**Figure 3 insects-12-00111-f003:**
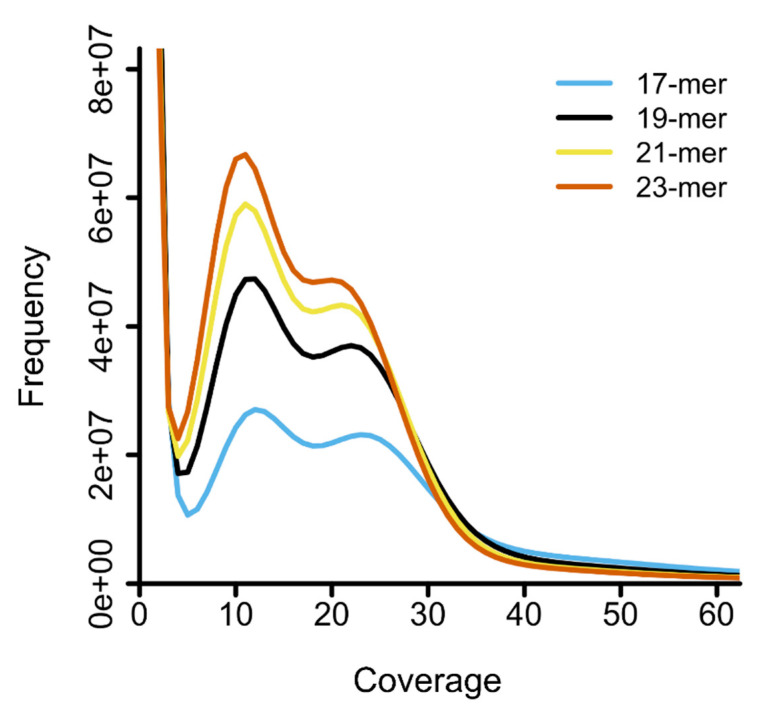
Distribution of four different *k*-mers by Jellyfish. The *X*-axis represents the sequencing coverage and the *Y*-axis indicates the frequency of each coverage. Each graph represents distribution from 17-mer to 23-mer by adding 2-mers to each *k*-mer. The major peak of each distribution was found at coverage 12 (17-mer and 19-mer) and 11 (21-mer and 23-mer), respectively.

**Figure 4 insects-12-00111-f004:**
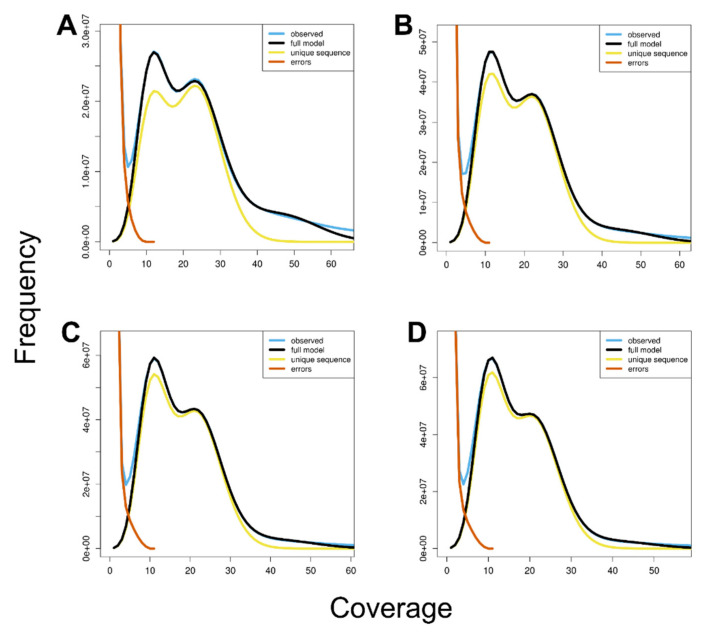
GenomeScope profiles of four different *k*-mers. The *X*-axis represents the sequencing coverage and the *Y*-axis indicates the coverage frequency. Each graph shows goodness of fit of the full model (black) with each observed *k*-mer frequency (blue). (**A**) 17-mer profile of GenomeScope; (**B**) 19-mer profile; (**C**) 21-mer profile; and (**D**) 23-mer profile.

**Table 1 insects-12-00111-t001:** Statistics on total reads of the *C. relictus* DNA-seq.

Insert size (bp)	550
Total raw reads	596,389,120
Total raw sequences (bp)	60,235,301,120
Average length of raw reads (bp)	101
Total trimmed reads	561,403,836
Total trimmed sequences (bp)	55,811,494,971
Average length of trimmed reads (bp)	99.4
Reads filtered out (%)	5.87
Sequences filtered out (%)	7.34

**Table 2 insects-12-00111-t002:** Genome sizes estimated by two tools at different *k*-mers.

Contents	17-mer	19-mer	21-mer	23-mer
Estimated genome size (bp) *	1,750,660,507	1,786,300,918	1,831,421,762	1,882,948,731
Estimated genome size **				
Heterozygosity (%)	1.71	1.81	1.77	1.70
Genome haploid length (bp)	1,517,383,829	1,562,628,029	1,589,446,896	1,611,360,578
Genome repeat length (bp)	1,096,296,131	883,450,205	796,477,101	757,352,211
Genome unique length (bp)	421,087,698	679,177,824	792,969,795	854,008,367
Model fit (%)	98.46	99.46	99.47	99.39
Read Error Rate (%)	0.15	0.23	0.25	0.25

* is a result carried out by Jellyfish. ** are results carried out by Genome Scope.

## Data Availability

The raw sequencing data are deposited in the NCBI Sequence Read Archive (SRA) database with accession PRJNA689978.

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
