# Peer review of "Genome Size Estimation of Callipogon relictus Semenov (Coleoptera: Cerambycidae), an Endangered Species and a Korea Natural Monument"

_insects, 2021, doi:10.3390/insects12020111_

Round 1

Reviewer 1 Report

The manuscript entitled “Genome size estimation of Callipogon relictus Semenov (Coleoptera: Cerambycidae), an endangered species and a Korea natural monument” offers genome size estimation data for a species of longhorn beetle. The manuscript is well written, and the results are clearly presented. Although the report is very brief since the authors only report genome size estimates for a single species, these data are rather scarce in Coleoptera, and therefore valuable to the growing genomics efforts in beetles.

Although I consider the manuscript and results to be generally sound, the reproducibility of the results is compromised by an incomplete description of the methods and, more importantly, by the lack of publicly accessible DNA sequencing data. Data availability in a publicly accessible database such as SRA or ENA is absolutely crucial, and I cannot recommend publication until this criterion is fulfilled.

As far as incomplete method description, please provide more details regarding the samples used in the paper, e.g. How many individuals were used in each experiment? Were the larvae sexed? If so, what was the sex of the individuals used in each experiment, and which tissues were used? Even if some of these details are presented later in the manuscript, they should also be described in detail the methods section.

Please provide the genome size estimate results for GenomeScope in Table 2. GenomeScope has its own model for estimating genome size, and since the authors are using results from this model (i.e. repetitive DNA content and heterozygosity) they should also show the genome size estimates generated by GenomeScope. Jellyfish results can be included in addition to GenomeScope.

I believe these major points can be addressed rather easily, and if the authors chose to do so I would believe the manuscript to be suitable for publication in Insects.

Minor points:

Lines 175 and 176. I believe you should replace “heterogeneity” and “homogeneity” with “heterozygous” and “homozygous”, respectively.

Author Response

The manuscript entitled “Genome size estimation of Callipogon relictus Semenov (Coleoptera: Cerambycidae), an endangered species and a Korea natural monument” offers genome size estimation data for a species of longhorn beetle. The manuscript is well written, and the results are clearly presented. Although the report is very brief since the authors only report genome size estimates for a single species, these data are rather scarce in Coleoptera, and therefore valuable to the growing genomics efforts in beetles.

Although I consider the manuscript and results to be generally sound, the reproducibility of the results is compromised by an incomplete description of the methods and, more importantly, by the lack of publicly accessible DNA sequencing data. Data availability in a publicly accessible database such as SRA or ENA is absolutely crucial, and I cannot recommend publication until this criterion is fulfilled.

Response:

First of all, we are grateful that the reviewer finds this work interesting.

All request data were uploaded to SRA database (project accession PRJNA689978). 

As far as incomplete method description, please provide more details regarding the samples used in the paper, e.g. How many individuals were used in each experiment? Were the larvae sexed? If so, what was the sex of the individuals used in each experiment, and which tissues were used? Even if some of these details are presented later in the manuscript, they should also be described in detail the methods section.

Response:

We greatly appreciate your valuable suggestions and comments for improving our manuscript.

Until now, it has not been reported and identified sex of the individuals in the larvae stage in Callipogon relictus species. Larvae of this species can’t be sexed at the larval stage. In most cases, the sexual identification of Cerambycidae can be done when the individual is at pupal stage.

The sentence has been added and modified to methods section as suggested. 

Please provide the genome size estimate results for GenomeScope in Table 2. GenomeScope has its own model for estimating genome size, and since the authors are using results from this model (i.e. repetitive DNA content and heterozygosity) they should also show the genome size estimates generated by GenomeScope. Jellyfish results can be included in addition to GenomeScope.

Response:

We appreciate for your important comments and correction in this data.

To clarify each result of genome size estimated by two tools, we have reformed the Table 2. In addition, the Jellyfish tool has provided the result of only estimated genome size.

Lines 175 and 176. I believe you should replace “heterogeneity” and “homogeneity” with “heterozygous” and “homozygous”, respectively 

Response:

We have replaced.

Reviewer 2 Report

The manuscript by Yu et al. " Genome size estimation of Callipogon relictus Semenov (Coleoptera: Cerambycidae), an endangered species and a Korea natural monument" performed the genome size estimation using flow cytometry and k-mer analysis to Callipogon relictus Semenov (endangered species). It is the first report about Genome size estimation in Callipogon relictus. It is thought to be worthy and expected to be utilized in further studies.
The aim and execution of the study are clear and in line. Such study and its results will indeed be a valuable resource for further study underlying Genome Project of Callipogon relictus. The techniques are appropriate and the illustrations for the most part good. The data documented in this manuscript should be useful to the community.

Minor comment
-English must be improved to ensure that all audiences can clearly understand the context of this manuscript. I advise the authors to get this manuscript checked by a native speaker who could help improve grammar and clarification of some text within your manuscript.

-The author should check italic and typo in the manuscript.

Author Response

English must be improved to ensure that all audiences can clearly understand the context of this manuscript. I advise the authors to get this manuscript checked by a native speaker who could help improve grammar and clarification of some text within your manuscript.

Response:

Thank you for your comments and suggestions!

According to the reviewer’s comment, we carefully rechecked the manuscript for any syntax errors in English and the revised manuscript has been crosschecked by a professional editing service.

The author should check italic and typo in the manuscript.

Response:

Has been modified.

Round 2

Reviewer 1 Report

The authors have responded to my comments and suggestions in full, and have uploaded the raw data associated with this study to the SRA. Thus, I am happy to recommend this manuscript for publication in Insects.